# Study of Energy Dissipation in the Mixed-State YBa_2_Cu_3_O_7-__δ_ Superconductor with Partially Deoxygenated Structures

**DOI:** 10.3390/ma15124260

**Published:** 2022-06-16

**Authors:** Artūras Jukna

**Affiliations:** Photovoltaic Technologies Laboratory, Vilnius Gediminas Technical University, Sauletekio av. 3, LT-10257 Vilnius, Lithuania; arturas.jukna@vilniustech.lt; Tel.: +370-5-251-2462

**Keywords:** type II superconductor, oxygen stoichiometry, pinning force, vortex motion

## Abstract

Energy dissipation from vortex motion, which appears as a resistivity of the mixed-state superconductor, limits the range of type II superconductors in low- and high-power electronics and optoelectronics. The level of dissipation increases with the development of the vortex motion phase from that of the thermally activated flux flow to that of the flux creep and finally to that of the flux flow. The vortex motion regimes depend on the balance between bias current-self-produced Lorentz force, accelerating vortices, and the pinning force, which, together with a magnetic drag force from pinned vortices, tends to stop the vortex motion. The current paper reports on energy dissipation in YBa_2_Cu_3_O_7-δ_ (YBCO) devices provided with partially deoxygenated structures mutually interacting by magnetic force with one another. The shape of the structure and the magnetic interaction between the trapped and moving vortices, as well as the magnetic interaction between neighboring structures, can cause the appearance of voltage steps in the device’s current–voltage characteristics observed in temperature range 0.94 ≥ *T*/*T*_c_ ≥ 0.98 (here, *T*_c_ = 91.4 K is the temperature of the superconducting transition in the YBCO material). Current findings demonstrate the potential of artificial structures to control vortex motion in a mixed-state YBCO superconductor by means of a temperature, bias current, and a specific configuration of the structure itself and a profile of the oxygen distribution in it.

## 1. Introduction

Mixed-state resistivity in a type II superconductor thin films biased with current *I* > *I*_c_ is related to a motion of Abrikosov vortices, which create the current-self-produced magnetic field penetrating the superconductor [1]. The number of magnetic vortices in the mixed-state superconductor increases with increasing current self-produced magnetic field inductance *B*, which, together with the current self-produced electric field, creates a Lorentz force *F*_L_ under which vortices nucleate at the film’s edges, move toward its center, and annihilate it there with antivortices approaching from the opposite edge of the film. The motion of vortices experiences three different phases closely related to the ratio between the *F*_L_, which depends on the amplitude of the bias current *I*, and the pinning force *F*_p_, which depends on the origin of the pinning centers in the superconductor [2,3]. These regimes are thermally activated flux flow (TAFF) observed when overall *F*_p_ > *F*_L_ and *I* < *I*_c_, flux creep (FC) when *F*_p_–*F*_L_ and *I*–*I*_c_ and flux flow when *F*_p_ < *F*_L_ and *I* > *I*_c_. Here, *I*_c_ is the superconducting critical current of the superconductor. While vortices pinned for crystal defects (such as point defects, vacancies [4], dislocations, columnar defects, twin boundaries, intergrowths, nanoparticles, voids, or local strained regions [2,3], etc.) of the superconductor thin film, the current flows only through the superconducting parts avoiding the normal state cores of the vortices [5]. The sample stays in the superconducting state and the current does not experience dissipation (i.e., *I* < *I*_c_). During the TAFF (random nucleation/motion of single vortices), FC (random nucleation/motion of bundles of vortices), or FF regimes (easy vortex motion), the current flows through the normal-state cores of moving vortices. Energy dissipation in experiments is expressed as the mixed-state resistance of a superconductor, which is proportional to the concentration and velocity of the moving vortices [5].

Periodic pinning structures [6] control the phase of flux flow [5] or even the phase of coherent motion of vortices in comparatively wide superconducting devices, i.e., with a width larger than the effective penetration depth of the magnetic field. The latter can observe through a step-like *I–V* characteristic of the mixed-state superconductor, similar to those of the Shapiro steps observed in microwave-irradiated [7] and in the absence of microwave radiation [8] superconducting devices. The *I–V* characteristic then obtains a linear character, showing the FF resistivity, and steps appear at voltages at which the inverse of the vortex time of flight across a superconductor strip coincides with the frequency of vortex/antivortex nucleation at the edges of the superconducting device [9].

When magnetic vortices are set in motion, the balance between the *F*_L_ and *F*_p_ forces varies with the magnetic viscous force [10], which increases linearly with increasing vortex velocity. Moving vortices can transfer energy to the magnetic moments in the magnetic superconductor by emitting spin waves. This behavior can contribute to the mass increase of the moving vortices. A magnetostatic interaction and a repulsion force between moving and stationary (i.e., strongly pinned) vortices can also contribute to the mass increase of moving vortices. In experiments, it can manifest itself as an overall increase in pinning force [10] for strongly interacting bundles of vortices that self-organize in the triangle (or hexagonal) magnetic lattice [2] known as the Abrikosov lattice. The magnetic drag on a moving magnetic vortex can arise from normal currents flowing in the core of the vortex line. The observed dissipation is believed to be related to the Magnus effect associated with a spinning object moving through a fluid and can be explained by eddy currents induced by the moving magnetic field of the vortex line [10,11].

Current work presents experimental data on the intermittency of flow of Abrikosov magnetic vortices in the YBa_2_Cu_3_O_7-δ_ (δ~0) superconducting thin film devices containing two partially deoxygenated (δ~0.2) structures for vortex motion. A magnetic interaction between vortices trapped on the slopes with vortices moving in the center of the structure is responsible for the appearance of voltage steps in the device’s current–voltage characteristics observed in a narrow temperature range close to *T*_c_ of the superconductor. This work demonstrates that the pinning force in the superconducting device can be controlled not only by artificially introducing additional oxygen vacancies in this structure, but also by giving the structure a specific configuration/form.

## 2. Materials and Methods

The samples were manufactured using a laser patterning of the oxygen-rich *c*-axis textured YBCO (δ~0) films grown on a LaAlO_3_ substrate of surface 1 × 1 cm^2^ area by means of a metal-organic chemical vapor deposition (MOCVD) technique. The films were characterized by the critical temperature *T*_c_ = 91.4 K with Δ*T*_c_~0.4 K and extremely large superconducting critical current density *J*_c_~3 MA/cm^2^ at temperature 78 K, arising from one-dimensional linear defects (e.g., screw and edge dislocations) that play the role of efficient pinning centers responsible for the enhancement of *J*_c_ in thin films [12]. To minimize heating at contacts, the *J*_c_ of the as-deposited YBCO films was measured using a 7-ns pulsed current technique for film biasing and sampling oscilloscope for measurements of voltage drop across the film [13]. In all measurement cases, the critical current was estimated as the current amplitude, which creates 10 μV voltage in the sample.

The 50 μm wide and 100 μm long YBCO stripes (samples) each insulated from one another and provided with 2 × 1.5 mm^2^ contact pads (Figure 1b–d) for Au/Cr electrodes were patterned in a nitrogen gas environment using a green light (λ = 532 nm) laser beam. The beam of laser fluence ~130 mJ/μm^2^ was focused in a Gaussian spot of ~5 μm in diameter and scanned at a speed of 5 μm/s over the stripe surface (white thick lines in Figure 1b–d) using a two-coordinate translation stage with the stripe fixed on it. The laser fluence was calculated as the optical energy of a laser divided by the optical spot area on the YBCO film surface with energy measurement performed at the film surface, i.e., replacing the film with an optical power meter. The diameter of the laser spot was determined using a microscope.

Joule heat due to light absorption-initiated oxygen diffusion from the illuminated areas of the film, resulting in a residual oxygen content δ > 0.6 and completely ruining superconductivity and ensuring electrical isolation of the test sample from the rest of the superconducting film. The reference sample (Figure 1b) was used for reference purposes.

Two partially deoxygenated lines (that is, straight lines and those that form rectangles) are manufactured by means of a laser beam of ~20 mJ/μm^2^ laser fluence and of 50 μm/s of scanning speed. Lower fluence and shorter exposure to a laser beam result in a partial depletion by oxygen (δ~0.2) of YBCO illuminated areas with YBCO; in this way, it does not ruin the superconductivity but creates additional oxygen vacancies, which play the role of very efficient pinning centers for magnetic vortices with an overall pinning force amplitude comparable to that of structural growth defects [4,14]. To increase the magnetic interaction between stationary and moving vortices, we introduced the second partially deoxygenated structure that forms a rectangle in samples 1 and 2. Both structures (straight and rectangle line) located at 10 μm distance in between their centers were prepared one by one using the same ~20 mJ/μm^2^ laser fluence and 50 μm/s scanning speed of the sample in a nitrogen gas. The main difference between samples 1 and 2 is the shape of a line that makes up a rectangle (Figure 1): sample 1 contains an oxygen-deoxygenated straight line and a line forming 40 μm × 30 μm rectangle, and sample 2 contains a straight line of partially deoxygenated material and a line forming a 10 μm × 30 μm rectangle. At temperatures below *T*_c_, the first magnetic field from the bias current penetrates the partially deoxygenated structures (regions of weak superconductivity) in the form of magnetic vortices and, when the Lorentz force exceeds the pinning force, the vortices start to move in both partially deoxygenated structures. Energy dissipation from moving vortices appears as a resistivity of the mixed-state superconductor, which has been investigated in a narrow range of temperatures ranging between 85.9 K (0.94·*T*_c_) and 89.6 K (0.98·*T*_c_).

The resistivity vs. temperature dependences and *I–V* characteristics of devices were investigated in a zero applied magnetic field using a standard four-probe measurement technique in the range of temperatures 80 K ≤ *T* ≤ 300 K when the sample was kept in a low vacuum in a cryostat. The ~0.1-μm-thick Au/Cr sandwich structures for contacts were deposited by the magnetron sputtering technique at room temperature, ensuring good adhesion to the film and negligible contact resistance, which was estimated using a three-probe measurement technique [13]. The setups for the resistivity vs. temperature dependences and *I–V* characteristics’ measurements were composed of a programmable current source connected to the outer Au/Cr contacts and a digital nanovoltmeter for voltage measurements connected to the inner contacts of the YBCO devices (Figure 1). The source of current and nano voltmeter was controlled with a computer program based on LabView.

## 3. Results and Discussion

### 3.1. Determination of Oxygen Distribution in the Partially Deoxygenated Structures

The optical energy absorbed in the YBCO material caused a local temperature rise and oxygen redistribution in the illuminated areas. The uneven distribution of optical power in the optical spot of Gaussian shape results in the appearance of a partially deoxygenated region with bevel slopes characterized by a relatively higher oxygen concentration compared to that in the central part of the structure (Figure 1). The residual oxygen content in the optically modified area was estimated as δ–0.2 using our results of measurements using a scanning electron microscope and energy dispersive (EDS) analysis of the chemical composition together with the results of the potential critical temperature *T*_c_ (estimated from resistivity vs. temperature dependences) and the room temperature resistivity of oxygen-deficient YBCO material using data in Refs. [15,16,17]. Because optically initiated heating, which causes a chemical diffusion of oxygen from the hot areas of the film toward the cold areas, depends on the heat conductivity in the illuminated area of the material, defects in the YBCO film can play a crucial role. As shown in [18,19], the diffusion of oxygen through grain boundaries in laser-untreated regions of the device can change the profile of the partially deoxygenated structure by producing regions rich and poor in the oxygen regions. The presence of the film substrate also complicates the situation, since the substrate blocks the optically initiated chemical diffusion of oxygen from the film, causing the formation of oxygen-enriched YBCO material near the YBCO/LaAlO_3_ interface. This means that the superconducting properties caused by a residual oxygen concentration in optically modified areas of the film can be different for material located on the top and bottom of the partially deoxygenated structure. In addition, inhomogeneous diffusion of oxygen forms a wavy oxygen profile on the slopes of the partially deoxygenated structure (Figure 1) with periodically recurring oxygen-enriched and partially deoxygenated areas. In summary, the results of the analysis in five different randomly selected places of the optically deoxygenated structures showed that the residual oxygen concentration on the slopes does not change monotonically as expected using a Gaussian-shaped optical spot for optical treatment of the YBCO film. Varying the oxygen concentration profile on the slopes of the partially deoxygenated structure, one can produce a device having a stepped-like current–voltage characteristic at temperatures close to the superconducting critical temperature of the optically deoxygenated structure of the YBCO material.

### 3.2. Resistivity versus Temperature Measurements

At temperatures *T* < *T*_c_ and biasing current *I* ≥ *I*_c_ of the superconductor, the central part of the partially deoxygenated structure and its bevel slopes are penetrated by a magnetic field of current in the form of Abrikosov magnetic vortices. Under the Lorentz force, the vortices and antivortices from opposite edges of the YBCO device start to move towards its center and annihilate there. The motion of vortices in the device causes the appearance of voltage, which is proportional to the number of moving vortices and their speed. The resistivity versus temperature dependences of samples 1 and 2 are shown in Figure 2a,b. Curve 1 in both graphs represents the resistivity vs. temperature dependence of the reference sample, which contains no artificially deoxygenated regions, and therefore the *T*_c_ value is the same as those in samples 1 and 2. The insertion of two partially deoxygenated structures in sample 1 and sample 2 causes an increase in their normal state resistivity by ~1.4 times and ~2.5 times, respectively, compared to the normal state resistivity of the reference sample. This difference in resistivity is caused by uneven distribution of current and electric field of current in sample 1 with partially deoxygenated structure shaped as a 40 μm × 30 μm rectangle and in sample 2 with a line forming a 10 μm × 30 μm rectangle. Due to the Meissner–Ochsenfeld effect at temperatures *T* < *T*_c_ and the presence of the 10 μm wide partially deoxygenated structure in the YBCO device, the current density in the central part of the device is higher compared to that of sample 2, in which the partially deoxygenated structure makes a 40 μm wide rectangle. Narrowing the rectangle, the bias current focuses on a central part of the device [20] (bottleneck), resulting in a larger amplitude of the electric field there than on the sidewalks of this rectangle (i.e., between the sides of the rectangle structure and the edges of the device (Figure 1)).

Due to vortex motion in partially deoxygenated regions, resistivity does not drop to zero but shows a repetitive recovery until ~50 μΩ cm in the biased sample 1 at temperatures *T* < 88 K (Figure 2a). Although at 0.8 μA bias current the resistivity drops to zero at temperatures below ~86 K, but with increasing current up to 1 μA, random recovering of resistivity takes place until *T* < 79 K.

It should be mentioned that quite often sample 1 obtains resistivity of ~25 μΩ⋅cm (Medium level). This result gives a clue that vortex motion takes place randomly in one of the partially deoxygenated structures or in both and depending on the number of moving vortices the dissipation expresses itself as sample resistivity of ~25 μΩ⋅cm or ~50 μΩ⋅cm if vortices are moving in one of these structures or in both structures, respectively.

The resistivity versus temperature dependence of sample 2 looks completely different from that measured for sample 1, although it also has two partially deoxygenated structures. A repetitive drop to zero and the recovery of resistivity is clearly seen only at bias current of 1 μA at temperatures 89.7 K< *T* < 90 K. The presence of a high (~50 μΩ⋅cm) and the medium levels resistivity (~25 μΩ⋅cm) in the resistivity vs. temperature dependence (Figure 2b) at temperatures *T* < 90 K confirms that vortex motion in sample 2 occurs either in one (medium level) or in both partially deoxygenated structures. The disappearance of the medium level resistivity at temperatures *T* < 89.7 K shows that, with decreasing temperature (or increasing the pinning force for vortices [6]) and increasing bias current (or increasing the Lorentz force for vortices), the vortex motion occurs in both deoxygenated structures. However, due to the pinning of vortices to the pinning centers in sample 2, the number of moving vortices in the structures decreases with decreasing temperature.

With increasing current up to 10 μA, the normal state room temperature resistivity of sample 2 increased by ~20 μΩ⋅cm. A similar increase in resistivity was observed in the mixed state at temperatures of 87.2 K < *T* < *T*_c_ (Figure 2b). However, when the bias current amplitude was set to 10 μA, the normal state and mixed state resistivity recovered reaching the same value as was measured for 1 μA current (curves 4 and 8 in Figure 2). The increase in resistivity with increasing current with later decrease back to an initial level could be associated with oxygen redistribution in the partially deoxygenated structure biased with current. The oxygen diffusion and drift through crystal 2D defects like those of grain boundaries and/or screw dislocations, which are characteristic in YBCO films grown by the MOCVD technique, can take place from partially deoxygenated regions towards regions rich in an oxygen, resulting in sample resistivity increase, and from regions rich in oxygen, toward regions poor in oxygen, resulting in resistivity decrease [18,19]. The chemical diffusion of oxygen (owing to the concentration gradient of oxygen), which causes the variation of the resistivity in the normal state, and the tracer diffusion (a spontaneous diffusion of oxygen in the absence of the concentration gradient), which causes the variation of the amplitude of the pinning force in the mixed state YBCO material, was investigated by many groups [18,19]. Assuming that the strongest electric field is expected in the vicinity of the bottleneck of the 10-μm-wide rectangle, the most intensive drift and diffusion of oxygen can take place in this area transporting the oxygen from one partially deoxygenated structure to another.

### 3.3. Analysis of Current–Voltage Characteristics

Due to an almost uniform pinning force in the entire partially deoxygenated structure, the vortex motion in samples 1 and 2 (curves 1 and 2) results in a remarkably higher level of energy dissipation than that of the reference sample (curve 3 in Figure 3), which is seen in the *I–V* characteristics measured at temperature *T* = 0.96·*T*_c_.

Due to the introduction of partially deoxygenated structures into YBCO devices, the critical current density *J*_c_~2 × 10^4^ A/cm^2^ measured in the reference sample decreased by more than three orders of magnitude to 9 A/cm^2^ and 2.7 A/cm^2^ in samples 1 and 2, respectively. In sample 2, the decrease in the critical current density is accompanied by the appearance of voltage steps (curve 2 in Figure 3). The stepped-like *I–V* characteristic was observed only in a narrow temperature range between 85.9 K (0.94·*T*_c_) and 89.6 K (0.98·*T*_c_) of YBCO and up to the 110th step (the resolution limit of the measurement setup). No voltage steps in the measured current range were observed in the *I–V* characteristics of sample 1 and the reference sample.

The stepped *I–V* characteristic is evidence that the self-produced Lorentz force of the bias current in the partially deoxygenated structure exceeds the pinning force and creates favorable conditions for vortex motion along the central part, which is characterized by the highest level of deoxygenation (or the highest concentration of oxygen vacancies). With increasing current, the self-produced magnetic field of the current penetrates not only the central part of the structure, but also its slopes, exhibiting a comparatively stronger pinning than that in the central part of this structure. Due to the magnetic interaction between the vortices trapped on the slopes and those moving in the center of the structure, the magnetic flux (each vortex captures the magnetic flux quantum Φ = *h*/(2*e*) ≈ 2.07 × 10^−15^ Wb) experiences a magnetic friction force [10,11]. This force tends to stop vortex motion resisting the Lorentz force and acts as an additional pinning for moving vortices. According to our estimations of oxygen distribution profile in random places of the partially deoxygenated structure (Figure 1), the slopes are not uniformly bevel and therefore the moving vortices appear to be confined and squeezed in a narrow central part of the structure. The magnetic interaction between moving and trapped vortices increases with increasing current, which can result in a random opening (the Lorentz force is greater than the pinning force) and closing (the Lorentz force is weaker than the pinning force) of the path for vortex motion.

It should be noted that stepped *I–V* characteristics were observed only in sample 2 (Figure 1d) containing a rectangle (10 μm × 30 μm) line structure. At *I* > *I*_c_, a rectangle of mixed state material produces a ~40 μm wide and ~10 μm wide bottleneck for the bias current in samples 1 and 2, respectively. The higher current density in the bottleneck results in a stronger magnetic field, which penetrates in the form of vortices/antivortices AB, BC, and CC segments (Figure 4) of the partially deoxygenated rectangle line and straight line (DD) structures. Therefore, the vortices appear to be more strongly squeezed in the central part of the structure, and, in the segment, CC ones can move only along a narrow pathway towards their annihilation line.

The opening of the path for the vortex motion experiences several phases. It starts with the thermally activated flux flow regime (TAFF), which is characterized by a low dynamic resistance (100–180 mΩ) estimated from the voltage step in the *I*–*V* characteristic (Figure 3) of sample 2 and associated with energy dissipation due to the random motion of individual vortices. With increasing current, the number of vortices trapped in the segments BC is increasing, and because of magnetic repulsion, the magnetic lattice of vortices in the section BC appears to be strongly squeezed, and when the magnetic repulsion force exceeds the pinning force, the bundles of vortices/antivortices enter the CC segment of the partially deoxygenated structure, move, and annihilate there (FC regime). Moving with velocity *v* vortices generates a voltage *V* = *nvH*. Here, *n* is the density of vortices and *H* is the strength of magnetic field captured in the partially deoxygenated structure. At fixed bias current of 1 μA, both structures are occupied with a randomly changing number of moving vortices as is seen in the resistivity versus temperature dependences (Figure 2), producing three-level dissipation in samples 1 and 2. Due to the different geometry and, therefore, different pinning conditions in both partially deoxygenated structures, the number of moving vortices is also different. Heat generated by moving and annihilating bundles of vortices/antivortices in the CC segment can affect the balance between *F*_L_ and *F*_p_ [21,22] and stop the movement of vortices. If so, the energy dissipation due to vortex motion will take place in the AB and BC segments of the structure, which are much longer than the CC segment in sample 2.

It should be mentioned that the absence of voltage steps in the *I–V* characteristic of sample 1 could be caused by several factors; first, the CC segment is almost four times longer than in sample 2. Its length is comparable with the length of 2(AB + BC), making a less significant density of current redistribution in the segments due to the change in the motion regime of vortices in the partially deoxygenated structure. Second, the extension of the CC section reduced the bottleneck for bias current. Due to the possible partially overlapping of AB and BC segments with sample edges, the conditions for vortex nucleation in rectangle-line and straight-line partially deoxygenated structures are almost the same, which can cause random nucleation of vortices either in one of the structures or simultaneously in both, as confirmed by three-level energy dissipation in Figure 2. Third, unequal pinning force conditions may occur in the longer CC segment, resulting in accidental vortex nucleation in random places of the CC segment. The factors mentioned above can potentially cause the step-free *I–V* characteristic of sample 1.

## 4. Conclusions

This work reports on experimental observations of energy dissipation biased with current mixed-state YBa_2_Cu_3_O_7-δ_ superconducting devices provided with optically deoxygenated (δ~0.2) structures, forming a bottleneck for the bias current. A proposed rectangle shape of the structure causes an increase in overall pinning for Abrikosov magnetic vortices, since some segments of this structure appear to be oriented perpendicularly to the Lorentz force. Pushing vortices perpendicularly to the current, the Lorentz force amplifies edge effect pinning in the structure. The experimental results demonstrate that, in the case of a narrow bottleneck (i.e., the rectangle structures segment embracing vortex-antivortex annihilation line is shorter than the rest of parallel to it segments of this structure) for the current, one can switch a regular *I–V* characteristic of the mixed state superconducting material to a stepped-like *I–V* characteristic. A turn-on/-off the vortex motion in segments of the partially deoxygenated structure affects current redistribution in the device, since current wanders the structure’s regions with the intensive vortex motion and flows through regions with slower motion of vortices that supports results reported in the literature. Understanding of this phenomenon provides a clue for developing new superconducting electronic devices based on vortex matter behavior in a mixed state superconductor, with a possibility to control device’s *I–V* characteristics/properties by means of a shape of the partially deoxygenated structure, the profile of oxygen distribution, and the deoxygenation level in the structure as well as by a biasing current.

## Figures and Tables

**Figure 1 materials-15-04260-f001:**
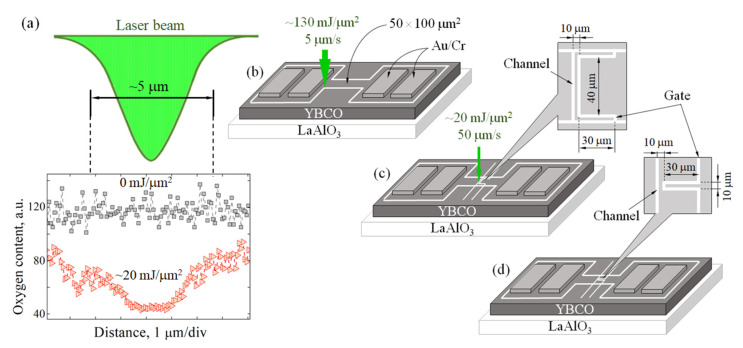
The partially deoxygenated (δ~0.2) structures in sample 1 (**c**) and sample 2 (**d**) are created by a laser beam, which produces laser fluence of ~20 mJ/μm^2^ on the sample surface, scanning it at a speed of 50 μm/s. The reference sample (**b**) does not have partially deoxygenated structures. About 5 μm wide insulating lines produced using the same laser with a laser fluence of 130 mJ/μm^2^ (white thick lines) form the outer contours of the samples. A typical profile of residual oxygen concentration in the film optically modified by the Gaussian laser beam of laser fluence ~20 mJ/μm^2^ is shown in (**a**). The upper curve of the graph represents the residual oxygen concentration in the optically untreated film area.

**Figure 2 materials-15-04260-f002:**
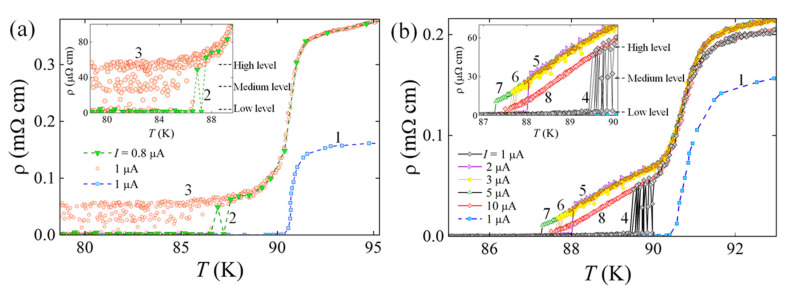
Resistivity versus temperature ρ(*T*) dependences of YBa_2_Cu_3_O_7-δ_ samples with partially deoxygenated structures (straight and rectangle lines) exhibiting the properties of a weak superconductor: (**a**) optically modified sample 1 biased with *I* = 0.8 μA (2) and 1 μA (3); (**b**) sample 2 biased with *I* = 1 μA (4), 2 μA (5), 3 μA (6), 5 μA (7), and 10 μA (8). Curve 1 in both graphs represents the same ρ(*T*) dependence of reference sample (having no partially deoxygenated lines) measured at *I* = 1 μA.

**Figure 3 materials-15-04260-f003:**
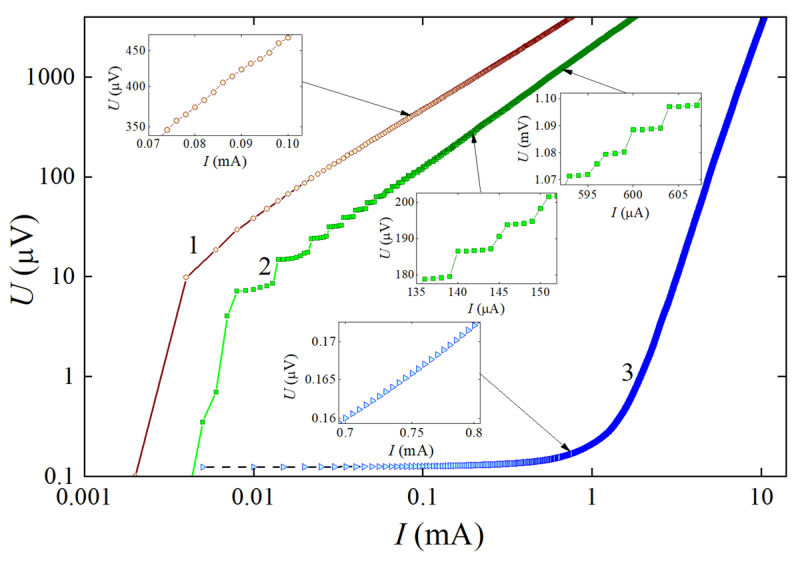
Log-log plot of the current–voltage (*I–V*) characteristics of sample 1 (curve 1), sample 2 (curve 2) and the reference sample (curve 3) measured at fixed temperature *T* = 87.74 K = 0.96·*T*_c_. The insets represent the fragments of the *V* characteristics on the expanded linear scale.

**Figure 4 materials-15-04260-f004:**
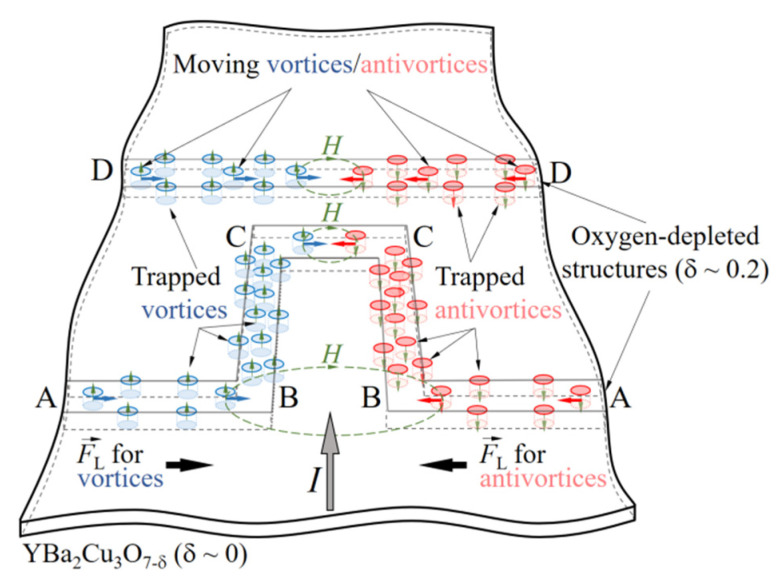
Vortices/antivortices motion scheme, demonstrating that the BC segments are oriented perpendicularly to the Lorentz force, which creates the edge effect pinning for vortices/antivortices there. If the Lorentz force exceeds the pinning force, the level of energy dissipation increases due to the onset of vortex motion in the CC segment of sample 2. When the vortices/antivortices stopped moving, the level of energy dissipation decreased.

## Data Availability

Not applicable.

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
