# Peer review of "Study of Energy Dissipation in the Mixed-State YBa2Cu3O7-δ Superconductor with Partially Deoxygenated Structures"

_materials, 2022, doi:10.3390/ma15124260_

Round 1

Reviewer 1 Report

The employment of type II superconductors in electronics is limited by the energy dissipation due to development of the vortex motion phase in the mixed-state of superconductor. The vortex motion regimes depend on the balance between the current-self-produced Lorentz force FL, accelerating vortices, and the pinning force FP, which tends to stop the vortex motion.

While vortices pined for crystal defects of the superconductor (FP > FL) the sample stays in the superconducting state and the current does not experience dissipation. Thus, an important task is to look for ways to regulate the effects of vortex pinning in superconducting systems.

The work of ArtÅ«ras JuknaStudy of energy dissipation in the mixed-state YBa2Cu3O7-d superconductor…” presents experimental data on YBa2Cu3O7-d (d ~ 0) superconducting thin film devices containing, created by a laser beam, partially oxygen-depleted (d ~ 0.2) structures. Resistivity versus temperature, r(T), dependences and the current-voltage, I-V, characteristics for two YBaCuO samples with different oxygen partially-depleted structures as well as r(T) and I-V dependences of the reference sample (having no oxygen-partially-depleted lines) are presented.

 Comparative analysis of experimental data displays that the pinning force in the superconducting device can be controlled not only by straightforward introducing of additional oxygen vacancies in the structure, but also by giving the structure a specific configuration.

The work demonstrates the potential of artificial structures to manage vortex motion in a mixed-state YBCO superconductor by means of a temperature, bias current, a specific configuration of the structure itself and by a profile of the oxygen distribution in it.

In particular, the intermittency of flow of magnetic vortices was found for one of the investigated thin film devices containing two oxygen-partially-depleted structures: the stepped I-V characteristic occurs due to the alternation of FP < FL and FP > FL regimes in the path for vortex motion as a consequence of specific geometry of depleted region.

In general, a formulation of the problem is clear, the task is well implemented, the results are interpreted in detail and the article of Artūras Jukna can be published in the presented form.

Author Response

Dear Reviewer,

Thank you for your remarks and the review of the manuscript.

Artūras Jukna

Reviewer 2 Report

The author finds steps in the current-voltage (I-V) characteristic of a YBCO thin film patterned with an oxygen-depleted track in the shape of a "U" upside down.  It is shown in Fig. 4: ABCC'B'A'.  This phenomenon occurs only at temperatures close to the critical temperature for superconductivity, Tc.  The author argues that the vortex lines are better pinned in the sections BC and B'C' of the track, and hence that electric current prefers to flow there, thereby creating a bottleneck.  He also argues that the steps that he observes in the I-V characteristic are due to the annihilation of vortices with anti-vortices at the CC' part of the track of oxygen depletion.

The author's results are very interesting, and they merit publication.  I suggest, however, that he answer the following questions first.

1. The author observes no voltage steps in the I-V characteristic in a sample that is patterned with a wider upside-down "U" track.  Although he provides some explanation for this effect in the conclusions section, they should be improved and made clearer.

2. What are the magnitudes of the magnetic fields in the sample?  They are are, of course, self fields.  See, for example, Talantsev and Tallon, Nature Communications 6, 7820 (2015).

3. Why do the current steps in the I-V characteristic appear only in a narrow region in temperature near Tc?  Could that be due to the decoupling of the vortex lines into "pancake" vortices inside the vortex-liquid phase? See, for example, Rodriguez, Phys. Rev. B 69, 100503(R), (2004); ?Phys. Rev. B 70, 224507 (2004).

4. Typograhical Errors:

Line 39, "pined" is spelled wrong

Line 280, replace "rectangular" with "rectangle"

volum6, Article number: 7820 (2015)

Author Response

Dear Reviewer,

Thank you for your remarks and the review of the manuscript.

Artūras Jukna

Please find below my answers to your questions/statements:

Question/Statement: The author observes no voltage steps in the I-V characteristic in a sample that is patterned with a wider upside-down "U" track.  Although he provides some explanation for this effect in the conclusions section, they should be improved and made clearer.

Answer: The explanation for the effect of no steps in the I-V dependence in a sample was extended and now is present in the discussion part of the manuscript:

It should be mentioned that the absence of voltage steps in the I-V characteristic of sample 1 could be caused by several factors; first, the CC segment is almost 4 times longer than in sample 2. Its length is comparable with the length of 2(AB + BC), making a less significant density of current redistribution in the segments due to the change in the motion regime of vortices in the oxygen-partially depleted structure. Second, the extension of the CC section reduced the bottleneck for bias current. Due to the possible partially over-lapping of AB and BC segments with sample edges, the conditions for vortex nucleation in rectangle-line and straight-line oxygen-depleted structures are almost the same, which can cause random nucleation of vortices either in one structure or simultaneously in both, as confirmed by three-level energy dissipation in Fig. 2. Third, unequal pinning force conditions may occur in the longer CC segment, resulting in accidental vortex nucleation in random places of the CC segment. The factors mentioned above can potentially cause the step-free I-V characteristic of sample 1.

Question/Statement: What are the magnitudes of the magnetic fields in the sample?  They are, of course, self-fields.  See, for example, Talantsev and Tallon, Nature Communications 6, 7820 (2015).

Answer: That is true. The origin of the magnetic field in the samples is the self-produced magnetic field of current. The magnetic field amplitude depends only on the strength of the bias current if one does not take into account the magnetic field of the Earth, which was present during these experiments. According to the reference by Talantsev and Tallon, Nature Communications 6, 7820 (2015), the strength of the magnetic field at bias current 10 uA in the homogenous 0.3 um thick and 50 um wide YBCO sample and with the London penetration depth of the order of 135 nm, the strength of the magnetic field would be 0.09 A/m or 0.113 uT. However, this would be valid only for homogeneous current flow through the whole cross-sectional area of the mixed-state sample. In the current case, Sample 2 contains an upside-down ‘U’ track of partially deoxygenated material, creating a bottleneck for the bias current that focuses on the center of the sample (Abrikosov vortices move along the CC sector). If vortices stop moving in sector CC, the current is redistributed to sectors AB and BC. Assuming that magnetic vortices can move only perpendicularly to the current direction, this variation in local current density changes the concentration of magnetic field lines penetrating through the partially deoxygenated material (i.e. the concentration of Abrikosov magnetic vortices) and the strength of the Lorentz force. Thus, in the current case, most likely, estimating the strength of the magnetic field in the local places of the sample is rather complicated.

No changes in the text of the manuscript were done.

Question/Statement: Why do the current steps in the I-V characteristic appear only in a narrow region in temperature near Tc?  Could that be due to the decoupling of the vortex lines into "pancake" vortices inside the vortex-liquid phase? See, for example, Rodriguez, Phys. Rev. B 69, 100503(R), (2004); ?Phys. Rev. B 70, 224507 (2004).

Stepped I-V characteristics were observed only in a narrow temperature range of temperatures ranging between 0.94·Tc and 0.98·Tc. The additional oxygen vacancies (point defects) introduced into the partially deoxygenated region of the device increase the pinning force for Abrikosov magnetic vortices, making it almost equal to that produced by structural defects, which are characteristic for YBCO films manufactured by means of the metal-organic chemical vapor deposition technique. It is believed that when the Lorentz force is greater than the pinning force, the vortices can freely move along the central part of the partially deoxygenated region, oriented perpendicularly to the bias current direction and independent of the vortex matter phase. The pinning force is very strong for Abrikosov vortices appearing in partially deoxygenated regions oriented parallel to the bias current. This makes a difference in vortex motion in the BC and CC segments of the partially deoxygenated region. 

The reason why the stepped I-V characteristic appears only in a narrow range of temperatures close to Tc is not sufficiently clear. Of course, one can speculate that most likely this temperature range is related to a smooth change in the ratio between the force amplitude of two sources: the first is a pinning force produced by the structural defects (grain boundaries and/or screw dislocations, which are characteristic defects in MOCVD prepared YBCO films) and the second one is produced by oxygen vacancies introduced during the deoxygenation process in this region. With decreasing temperature, the pinning force produced by the structural defects may increase more rapidly than that produced by the oxygen vacancies. It ruins the uniformity of the pinning force causing an increase in overall pinning in the partially deoxygenated region. Then, nucleation of vortices appears randomly in random places in the partially deoxygenated region, and the sample behaves like a regular superconducting film biased at the supercritical current. However, to prove it, more experiments with different geometry of the upside-down ‘U’ track are needed.   

No changes in the text of the manuscript were done.

  1. Typographical Errors:

Line 39, "pined" is spelled wrong

Line 280, replace "rectangular" with "rectangle"

Answer: Typographical errors corrected in the text of the manuscript. Thank you. The corrections are shown in yellow in an updated version of the manuscript.

Reviewer 3 Report

Jukna presentes an experimental study on the vortex montion in high-Tc superconducting film, where the structure is modified by partial deoxygenation.

The Introduction provides background on vortex motion in superconductors, but it overlooks the very important recent development of electric field (gate-voltage) modified critical current: electric field can either increase or decrease the critical current, such as

Rocci et al Nanolett. (2021) 21, 1, 216-221 "Large enhancement of critical current in superconducting devices by gate voltage"

de Simoni et al Nature Nanotech. (2018) 13, 802-805 "Metallic supercurrent field-effect transistor"

The experimental methods are not described adequately.

How was the YBCO film grown and characterized?

The device geometry, that is to say the shape of the oxygen depleted regions in the two samples, is not described in an easy to understand manner, either in Fig. 1 or the corresponding text. I recommend using some shades for the affected regions, instead of surrounding them by white lines (unless it is the white lines themselves that are the irradiated regions, goes to show I'm unclear what happened), and replace the simple term "rectangles" (certainly not rectangulars!) in the text with something that better describes whatever happened.

How was the resistivity and critical current determined? What instrumentation and measurement scheme were used?

The starting sentence in line 141 either misses the statement or is a jumble of several phrases, I can't make out its meaning. There are many such sentences.

Was the oxygen content characterized by EDX? Is that sufficiently sensitive to oxygen to characterize the depletion to the indicated precision? This part should also be described in the Methods section, not just in the Results.

The discussion refers to sample RS but other parts of the text refer to it as "reference sample" or similar.

Section 3.2 seems to imply that 10uA bias current caused oxygen diffusion. If that were the case, then a posterior measurement at lower bias would also show smaller resistances.

Section 3.3 start by referring to Samples 1, 2 and 3, but the rest of the paper defined Samples 1 and as those affected by irradiation and also mentioned the reference, or sometimes RS, sample. This must be made uniform and intuitive throughout the manuscript.

The definition of the critical current measurement (10 uV) is only mentioned hidden in Section 3.3, it should have been upfront in Methods.

Caption 3 mentions "the" V characteristic, presumably I-V? and repeated in the text (ie. line 279).

Caption 4 misses large segments of a phrase, its meaning can't be deciphered.

The most important discussion starts around line 295 and it refers in detail to the structures of the two itrradiated samples, the relative lengths of the lines etc. However, this is difficult to follow since the strucrures were inadequately described around Fig. 1.

How would the heat generated by the moving and annihilating vortices stop their movement when the pinning force decreases at higher temperatures?

Overall, although the experiment shows indications of modified vortex dynamics based on the geometry of bottlenecks created by deoxygenated regions, due to the hard-to-control nature of oxygen defects, as described in the manuscript (inhomogeneous transversally across the film thockness and longitudinally across the irradiated regions) it appears to be difficult to draw strong conclusions from the data gathered on a limited number of structures. 

English: although the text has clearly been edited, it does contain typoes such as pining/pined instead of pinning/pinned at several spots, or missing verbs and words, or unfinished sentences, for example line 52-53, or repeated phrases such as "using a green light laser beam of green light", or wrong word order such as oxygen partially depleted, or mixed grammar like "stronger squeezed" instead of more strongly squeezed. The text needs to be carefully revised again for such sloppy editing errors, I'm not going to list more of them. 

Author Response

Dear Reviewer,

Thank you for your remarks and the review of the manuscript.

Artūras Jukna

Please find below my answers to your questions/statements:

Question/Statement: The Introduction provides background on vortex motion in superconductors, but it overlooks the very important recent development of electric field (gate-voltage) modified critical current: electric field can either increase or decrease the critical current, such as Rocci et al Nanolett. (2021) 21, 1, 216-221 "Large enhancement of critical current in superconducting devices by gate voltage" de Simoni et al Nature Nanotech. (2018) 13, 802-805 "Metallic supercurrent field-effect transistor"

Answer: The critical current in our devices was measured using a 10 uV voltage criterion, i.e. this voltage magnitude develops a bias current flowing between outer contacts while voltage is measured between inner contacts (as it is shown in Fig. 1 of the current manuscript). As I mentioned in the manuscript, the thickness of the YBCO sample is 300 nm, and the length is 100 um. These dimensions are almost two orders in magnitude larger than the ones mentioned in the paper by Rocci et al. Nanolett. (2021) 21, 1, 216-221 "Large enhancement of critical current in superconducting devices by gate voltage".  If we compare the thickness of YBCO film and the coherence length measured in the YBCO material, the thickness is two orders in magnitude larger than that of the thickness of the NbN film device investigated by Rocci et al. Nanolett. (2021) 21, 1, 216-221. The coherence length in the NdN is of the same order as film thickness. Therefore, please let me disagree with your opinion that in the YBCO samples, I used for my experiments, the enhancement of critical current by gate voltage can take place.

No changes in the text of the manuscript were done.

Question/statement: The experimental methods are not described adequately. How was the YBCO film grown and characterized?

Answer: text describing the characterization of YBCO films added in rows 88-90:

‘To minimize heating at contacts, the Jc of the as-deposited YBCO films was measured using a 7-ns pulsed current technique for film biasing and sampling oscilloscope for measurements of the voltage drop across the film [14]. ‘

Reference [15] was substituted for reference [14] according to its appearance in the text of the manuscript.

Question/Statement: The device geometry, that is to say the shape of the oxygen depleted regions in the two samples, is not described in an easy to understand manner, either in Fig. 1 or the corresponding text. I recommend using some shades for the affected regions, instead of surrounding them by white lines (unless it is the white lines themselves that are the irradiated regions, goes to show I'm unclear what happened), and replace the simple term "rectangles" (certainly not rectangulars!) in the text with something that better describes whatever happened.

Answer: The caption to figure 1 is updated with extra text:

Figure 1. … A ~5 mm wide insulating lines produced using the same laser with a laser fluence of 130 mJ/mm2 (white thick lines) form the outer contours of the samples. ….

And in the text of the manuscript (rows 96-97):

…The beam of laser fluence ~130 mJ/mm2 was focused in a Gaussian spot of ~5 mm in diameter and scanned at a speed of 5 mm/s over the stripe surface (white thick lines in Fig. 1b, c, and d) using a two-coordinate translation stage with the stripe fixed on it.

All terms ‘rectangulars’ are replaced with ‘rectangles’.

Question/Statement: How was the resistivity and critical current determined? What instrumentation and measurement scheme were used?

Answer:

(In rows 139-140): ‘The resistivity vs. temperature dependences and I-V characteristics of devices were investigated using a standard four-probe measurement technique in the range of temperatures 80 K £ T £ 300 K when the sample was kept in a low vacuum in a cryostat. The ~0.1-mm-thick Au/Cr sandwich structures for contacts were deposited by the magnetron sputtering technique at room temperature, ensuring good adhesion to the film and negligible contact resistance, which was estimated using a three-probe measurement technique [14].’

and (in rows 145-147): ‘The setups for the resistivity vs. temperature dependences and I-V characteristics measurements were composed of a programmable current source for current supply through the outer Au/Cr contacts of the devices (Fig. 1) and measurements and a digital nanovoltmeter for voltage measurements across the devices using their inner Au/Cr contacts. The source of current and nanovoltmeter was controlled with a computer program based on LabView.’ 

Question/Statement: The starting sentence in line 141 either misses the statement or is a jumble of several phrases, I can't make out its meaning. There are many such sentences.

Answer: (A corrected text in rows 153-161):

‘The optical energy absorbed in the YBCO material caused a local temperature rise and oxygen redistribution in the illuminated areas. The uneven distribution of optical power in the optical spot of Gaussian shape results in the appearance of a partially deoxygenated region with bevel slopes characterized by a relatively higher oxygen concentration compared to that in the central part of the structure (Fig. 1). The residual oxygen content in the optically modified area was estimated as d ~ 0.2 using our results of measurements by means of a scanning electron microscope and energy dispersive (EDS) analysis of the chemical composition together with the results of potential critical temperature Tc (estimated from the resistivity vs. temperature dependences) and the room temperature resistivity of oxygen-deficient YBCO material using data in Refs […]’

with 3 new references added in the section References.

Question/Statement: Was the oxygen content characterized by EDX? Is that sufficiently sensitive to oxygen to characterize the depletion to the indicated precision? This part should also be described in the Methods section, not just in the Results.

Answer: That is true. The oxygen content was characterized by EDX. Of course, the sensitivity for oxygen characterization is not enough. EDX results allow estimating a profile of oxygen distribution in the partially deoxygenated area of the YBCO material, but not the absolute value of oxygen concentration. For determination of the average value of oxygen concentration in a partially deoxygenated region EDX results were used in parallel with results of Tc and room temperature measurements of other authors. The text was updated with this explanation:

The residual oxygen content in the optically modified area was estimated as d ~ 0.2 using our results of measurements by means of a scanning electron microscope and energy dispersive (EDS) analysis of the chemical composition together with the results of potential critical temperature Tc (estimated using the resistivity vs. temperature dependences) and the room temperature resistivity of oxygen-deficient YBCO material using data in Refs [16-18]’ Here they are

  1. Jones, E.C., Christen, D.K., Thompson, J.R., Feenstra, R., Zhu, S., Lowndes, D.H., Phillips, J.M., Siegal, M.P., Budai, J.D. Correlations between the Hall coefficient and the superconducting transport properties of oxygen-deficient YBa2Cu3O7-d epitaxial thin films. Rev B 1993, 47, 8986–8995.
  2. Cava, R.J., Batlogg, B., Chen, C.H., Rietman, E.A., Zahurak, S.M., Werder, D. Single-phase 60-K bulk superconductor in annealed Ba2YCu3O7-d (0.3 < d < 0.4) with correlated oxygen vacancies in the Cu-O chains. Rev B 1987, 36, 5719–5722.
  3. Cava, R.J., Hewat, A.W., Hewat, E.A., Batlogg, B., Marezio, M., Rabe, K.M., Krajewski, J.J., Peck Jr., W.F., Rupp Jr., L.W. Structural anomalies, oxygen ordering and superconductivity in oxygen deficient Ba2YCu3Ox. Physica C: Superconductivity 1990, 165, 419–433.  

Question/Statement: The discussion refers to sample RS but other parts of the text refer to it as "reference sample" or similar.

Answer: The abbreviation RS has been removed from the text of the manuscript.

Question/Statement: Section 3.2 seems to imply that 10 uA bias current caused oxygen diffusion. If that were the case, then a posterior measurement at lower bias would also show smaller resistances.

Answer: I think that diffusion of oxygen in samples with partially deoxygenated structures takes place always since samples 1 and 2 have oxygen-doped and depleted regions located one next to another. I believe that the drift of oxygen in YBCO material can take place at some biasing voltage amplitude resulting in a strong enough electric field in the sample at the boundary between differently doped regions. What is important is that at biasing currents of 1 uA and 10 uA we have the same shape of resistivity versus temperature dependence starting from the normal-state resistivity down to 89 K. At bias currents of 2 uA, 3 uA, and 5 uA, the resistivity is noticeably higher. This means that flow of current changes its pathway. This is why I think this change happens to be due to oxygen diffusion/drift in the sample.

No changes were made to the text of the manuscript.

Question/Statement: Section 3.3 start by referring to Samples 1, 2, and 3, but the rest of the paper defined Samples 1 and as those affected by irradiation and also mentioned the reference, or sometimes RS, sample. This must be made uniform and intuitive throughout the manuscript.

Answer: A mistyping has been removed from the text.

Question/Statement: The definition of the critical current measurement (10 uV) is only mentioned hidden in Section 3.3, it should have been upfront in Methods.

Answer:  sentence ‘In all measurement cases, the critical current was estimated as the current amplitude, which creates 10 mV voltage in the sample.’ moved to section 2 Material and methods.

Question/Statement: Caption 3 mentions "the" V characteristic, presumably I-V? and repeated in the text (ie. line 279).

Answer: The sentence was corrected. Now it is as follows ‘…Needs noting that stepped I-V characteristics were observed only in sample 2 (Fig. 1d) containing a rectangle (10 mm ´ 30 mm) line structure…’

Question/Statement: Caption 4 misses large segments of a phrase, its meaning can't be deciphered.

Answer: Caption 4 corrected in such a manner ‘Figure 4. Vortices/antivortices motion scheme, demonstrating that the BC segments are oriented perpendicularly to the Lorentz force, which creates the edge effect pinning for vortices/antivortices there. If the Lorentz force exceeds the pinning force, the level of energy dissipation increases due to the onset of vortex motion in the CC segment of sample 2. When the vortices/antivortices stopped moving, the level of energy dissipation decreased.’

Question/Statement: The most important discussion starts around line 295 and it refers in detail to the structures of the two itrradiated samples, the relative lengths of the lines etc. However, this is difficult to follow since the structures were inadequately described around Fig. 1.

Answer: Now figure 1 capture mentions the meaning of white lines.

‘A ~5 mm wide insulating lines produced using the same laser with a laser fluence of 130 mJ/mm2 (white thick lines) form the outer contours of the samples’

Question/Statement: How would the heat generated by the moving and annihilating vortices stop their movement when the pinning force decreases at higher temperatures?

Answer: While vortices/antivortices move in the segment CC of sample 2, the pinning there stays more or less uniform. To initiate the motion of vortices/antivortices, the Lorentz force should be slightly stronger than the pinning force in the segment CC. Due to excess heat from vortex motion/annihilation, the uniform pinning ruins and region of slightly higher temperature can initiate a redistribution of current in the rectangle structure in this way, changing the amplitude of the Lorentz force for vortices moving in the segment CC. A redistribution of current can establish a new balance between the Lorentz force and the CC pinning force in the segment of sample 2.

Question/Statement: Overall, although the experiment shows indications of modified vortex dynamics based on the geometry of bottlenecks created by deoxygenated regions, due to the hard-to-control nature of oxygen defects, as described in the manuscript (inhomogeneous transversally across the film thickness and longitudinally across the irradiated regions) it appears to be difficult to draw strong conclusions from the data gathered on a limited number of structures. 

Answer: The main conclusion is that this work demonstrates the potential of artificial structures to manage vortex motion in a mixed-state YBCO superconductor. It is shown in the manuscript that it could be reached by means of a temperature and bias current, by a geometry of the oxygen-depleted structure, and by a profile of the profile of oxygen concentration in it. To draw strong conclusions, one needs to optimize the device. This could be done on a larger number of structures and/or using a computer-based simulation. I think it can possibly be done in the future.

Question/Statement: English: although the text has clearly been edited, it does contain typoes such as pining/pined instead of pinning/pinned at several spots, or missing verbs and words, or unfinished sentences, for example, lines 52-53, or repeated phrases such as "using a green light laser beam of green light", or wrong word order such as oxygen partially depleted, or mixed grammar like "stronger squeezed" instead of more strongly squeezed. The text needs to be carefully revised again for such sloppy editing errors, I'm not going to list more of them. 

Answer: all mentioned mistyping or text errors are corrected and given in yellow in the text of the manuscript.

Round 2

Reviewer 3 Report

the manuscript may be accepted now